# SARS-CoV-2 Seroconversion in an Adult Horse with Direct Contact to a COVID-19 Individual

**DOI:** 10.3390/v14051047

**Published:** 2022-05-14

**Authors:** Nicola Pusterla, Antoine Chaillon, Caroline Ignacio, Davey M. Smith, Samantha Barnum, Kaila O. Y. Lawton, Greg Smith, Bradley Pickering

**Affiliations:** 1Department of Medicine and Epidemiology, School of Veterinary Medicine, University of California, Davis, CA 95616, USA; smmapes@ucdavis.edu (S.B.); kolawton@ucdavis.edu (K.O.Y.L.); 2Division of Infectious Diseases and Global Public Health, University of California San Diego, La Jolla, CA 92093, USA; achaillon@health.ucsd.edu (A.C.); cignacio@health.ucsd.edu (C.I.); d13smith@health.ucsd.edu (D.M.S.); 3National Center for Foreign Animal Disease, Canadian Food Inspection Agency, Winnipeg, MB R3E 3M4, Canada; greg.smith@inspection.gc.ca (G.S.); bradley.pickering@inspection.gc.ca (B.P.)

**Keywords:** SARS-CoV-2, horse, antibodies, COVID-19, spillover

## Abstract

The authors report on a possible direct exposure to SARS-CoV-2 from a COVID-19-positive individual to an adult horse. The individual, diagnosed with COVID-19 (Delta B.1.617.2), had daily contact to her two horses prior to and during the development of clinical disease. None of the two horses developed abnormal clinical signs or had detectable SARS-CoV-2 in blood, nasal secretion, or feces via RT-qPCR. However, one of the two horses showed close temporal seroconversion to SARS-CoV-2 using a protein-based ELISA and the plaque reduction neutralization test. The results suggest that horses can become silently infected with SARS-CoV-2 following close contact with humans infected with SARS-CoV-2. As a precautionary measure, humans infected with SARS-CoV-2 should avoid close contact with equids and other companion animals during the time of their illness to prevent viral transmission.

## 1. Introduction

Susceptibility and exposure to SARS-CoV-2 from COVID-19 patients are prerequisites for animals to become infected [1]. Natural transmission of SARS-CoV-2 from clinical and asymptomatic COVID-19 individuals has been suspected in dogs, cats, tigers, lions, and minks [2,3,4]. The role of equids in the COVID-19 pandemic has remained poorly investigated. Comparative analysis of ACE-2 protein sequences has shown that equids (horses and donkeys) have a low affinity to bind SARS-CoV-2 [5]. A recent study aimed at determining functional and genetic analysis of viral receptor ACE-2 orthologs showed that ACE-2 orthologs from 44 species, including horses, could bind the SARS-CoV-2 spike protein and support viral entry [6]. Experimental infection of a single adult horse using intranasally administered ancestral SARS-CoV-2 yielded no molecular detection of SARS-CoV-2 in nasal secretions or feces and no virus isolation from respiratory tissues [7]. These results are in line with previous experimental studies in horses using the closely related MERS-CoV, showing a lack of immune response and no viral RNA detected in the upper respiratory tract [8]. While experimental studies are needed to establish susceptibility and investigate risk of horse-to-horse and/or horse-to-human transmission, such protocols never reproduce natural conditions. Recent studies looking at the molecular detection of SARS-CoV-2 in nasal secretions and/or feces from healthy horses in contact with infected keepers and from horses with acute onset of fever and respiratory signs did not find evidence of SARS-CoV-2 shedding [9,10]. Seroepidemiological studies have been shown to be more accurate at documenting past exposure to SARS-CoV-2 in companion animals and livestock [11]. A recent serological survey from China, did not find antibodies specific to SARS-CoV-2 from 18 healthy horses [12]. This was in sharp contrast to a study evaluating silent transmission of SARS-CoV-2 between racetrack workers with asymptomatic COVID-19 and racing Thoroughbred horses [10]. The latter study showed that 5.9% of healthy racehorses had detectable antibodies to SARS-CoV-2, suggesting possible spillover from COVID-19 humans to horses. Here, we report on the seroconversion to SARS-CoV-2 in an adult horse with direct contact to a COVID-19-positive individual. To the authors’ knowledge, this is the first report linking exposure from a human with COVD-19 to a susceptible horse.

## 2. Materials and Methods

In October of 2021, a middle-aged woman developed fever, fatigue, myalgia, and anosmia. Shortly after developing symptoms, the individual tested positive for SARS-CoV-2 via a rapid antigen-test and a confirmatory RT-qPCR test. Up to the development of clinical signs and following their rapid resolution, the woman cared for her 8- and 21-year-old Quarter horse mares on a daily basis. The approximate 2 h daily care consisted of the twice-daily feeding, stall cleaning, grooming, and exercising. Due to the woman’s concern for possible exposure of her two horses to SARS-CoV-2, one of the authors (N.P.) was contacted in order to monitor the horses on a regular basis. Horses were monitored daily via routine physical assessment and blood, nasal secretions, and feces were collected weekly for three weeks following the woman’s COVID-19 diagnosis. Additional blood samples were collected up to 60 days post-COVID-19 diagnosis. The various biological samples were tested for SARS-CoV-2 via RT-qPCR as previously described [10]. Weekly serum samples were tested for antibodies to SARS-CoV-2 using an ELISA targeting the receptor-binding domain (RBD) of the S protein. The cut-off value for the ELISA was determined as six times the standard deviation above the mean value of reactivity of 88 seronegative samples from a pre-COVID-19 cohort of healthy adult horses as previously reported [10]. The serum samples were also tested via the plaque reduction neutralization test (PRNT). As a baseline for neutralization with horse sera for SARS-CoV-2 has not yet been established for equids, the results were reported as >70% (PRNT_70_) and >90% (PRNT_90_) reduction in the number of plaques [13].

## 3. Results

Genotyping of SARS-CoV-2 from the horse owner was consistent with the temporally circulating Delta B.1.617.2 variant. SARS-CoV-2 full genome sequencing was performed on the participant positive sample using the ARTIC protocol and Illumina MiSeq platform [14]. Both horses remained healthy throughout the three-week monitoring period. Whole blood, nasal secretions, and feces collected at onset of COVID-19 diagnosis and 7, 14, and 21 days thereafter tested RT-qPCR-negative for SARS-CoV-2. Antibodies were detected only in the younger of the two mares with a rise in IgG against the RBD of the spike protein at 7 days post-COVID-19 diagnosis of the owner (Figure 1). Peak antibody titer was reached at 21 days post-COVID-19 diagnosis and remained elevated up to the last sample collection. The PRNT titers paralleled the ELISA titers and ranged from 5 to 20 for the PRNT_70_ and from 5 to 10 for the PRNT_90_.

## 4. Discussion

Despite the low susceptibility of equids to SARS-CoV-2 and little evidence for natural infection, this present case report describes a potential spillover from a COVID-19 patient to one of her two horses. Various companion animal species such as dogs and cats have shown little disease expression, despite being susceptible to SARS-CoV-2 [15]. Apparently, horses follow the same pattern after infection with SARS-CoV-2. This observation is also supported by the inability to detect SARS-CoV-2 by RT-qPCR in 667 horses with fever and respiratory signs [10]. The lack of SARS-CoV-2 detection via RT-qPCR in the horse that seroconverted likely related to the short shedding period and the weekly sample collection interval. A similar pattern comprising seroconversion with no to very short viremia and nasal shedding has recently been reported in dogs and pigs, which relate to the low susceptibility of these animal species to SARS-CoV-2 [15,16,17]. While various host, viral, and environmental factors can predispose infection, it was interesting to notice that only the younger of the two horses seroconverted to SARS-CoV-2. While the number of infected horses is too low to draw any age-related association, it is interesting to notice that younger cats are apparently more vulnerable than older ones [14]. Without the detection and characterization of SARS-CoV-2 from the horse, the source of infection remains speculative; however, the close temporal association between the COVID-19 individual and the antibody response of the horse suggest direct transmission from human to animal. While a previous attempt to experimentally infect a single horse with SARS-CoV-2 (virus strain 2019-nCoV/USA-WA1/2020) failed, it is possible that certain human-adapted variants are more likely to replicate in equids [7]. It will be interesting to determine if new variants (e.g., Omicron variant), known to have an increased rate of virus transmission, will be more likely to induce silent infection in equids [18]. The antibody response observed in the young Quarter horse mare was, compared to more susceptible animal species, low. This observation likely reflects a short infection phase, supported by the lack of clinical disease and absence of detectable virus. However, the antibody response persisted during the entire study period. Antibodies to SARS-CoV-2 have been shown to persist up to 10 months in some dogs and cats from COVID-19 positive households [19].

At the present time and based on the limited scientific data available, it appears that the overall risk of SARS-CoV-2 transmission between humans with COVID-19 and equids is low; however, it is important to continue to study the impact of potential spillover via longitudinal studies aimed at sampling horses at regular intervals once caretakers or horse owners have been diagnosed with COVID-19. It is imperative to determine the ability of SARS-CoV-2 to infect and become established in different animal species such as horses, especially in light of possible animal-to-human transmission [20]. While there is no evidence for horse-to-horse transmission of SARS-CoV-2, the current guidelines recommend that owners infected with SARS-CoV-2 avoid any close contact with their animals, including horses.

## Figures and Tables

**Figure 1 viruses-14-01047-f001:**
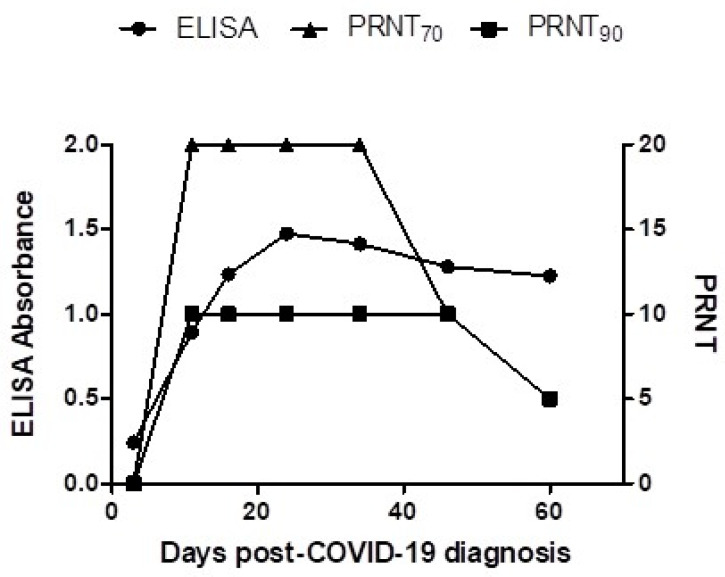
Antibody kinetics to SARS-CoV-2 in an 8-year-old Quarter horse following possible spillover infection from a COVID-19 individual. Antibodies were measured using an ELISA (circles), the PRNT_70_ (triangles), and the PRNT_90_ (squares).

## Data Availability

Not applicable.

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
