# Peer review of "SARS-CoV-2 Seroconversion in an Adult Horse with Direct Contact to a COVID-19 Individual"

_viruses, 2022, doi:10.3390/v14051047_

Round 1

Reviewer 1 Report

The case reported by Bradley et al. claimed a possible direct exposure to SARS-CoV-2 from a COVID-19 individual to an adult horse. Although no detectable SARS-Cov-2 RNA was founded by RT-qPCR, one of the two horses showed close temporal seroconversion to SARS-CoV-2 using a protein-based ELISA and the plaque reduction neutralization test. This is an interesting and meaningful report.

Major comments

It is possible that SARS-Cov-2 could infect horse based on the results the author provided. It would be perfect if these authors provide some evidences to confirm the infection in vitro using live virus or pseudovirus ? Just like this paper Functional and genetic analysis of viral receptor ACE2 orthologs reveals a broad potential host range of SARS-CoV-2 | PNAS

At least, the author need to propose some explanations and conjectures of this infection in discussion part. Does the viral receptor ACE of horse mediate the binding and infection of SARS-CoV-2?

Author Response

The case reported by Bradley et al. claimed a possible direct exposure to SARS-CoV-2 from a COVID-19 individual to an adult horse. Although no detectable SARS-Cov-2 RNA was founded by RT-qPCR, one of the two horses showed close temporal seroconversion to SARS-CoV-2 using a protein-based ELISA and the plaque reduction neutralization test. This is an interesting and meaningful report.

Major comments

It is possible that SARS-Cov-2 could infect horse based on the results the author provided. It would be perfect if these authors provide some evidences to confirm the infection in vitro using live virus or pseudovirus ? Just like this paper Functional and genetic analysis of viral receptor ACE2 orthologs reveals a broad potential host range of SARS-CoV-2 | PNAS

The authors agree with the reviewer that with absence of either molecular detection of SARS-CoV-2 or successful culture of SARS-CoV-2, the case report is speculative. However, a similar pattern of infection has been reported for less-susceptible animal species such as dogs and pigs, where only seroconversion was detected in the absence of viral detection in respiratory tissues. The reference mentioned by the reviewer (Liu, Y., Hu, G., Wang, Y., Ren, W., Zhao, X., Ji, F., Zhu, Y., Feng, F., Gong, M., Ju, X., Zhu, Y., Cai, X., Lan, J., Guo, J., Xie, M., Dong, L., Zhu, Z., Na, J., Wu, J., Lan, X., Xie, Y., Wang, X., Yuan, Z., Zhang, R., Ding, Q. Functional and genetic analysis of viral receptor ACE2 orthologs reveals a broad potential host range of SARS-CoV-2. Proc Natl Acad Sci U S A 2021, 118, e2025373118) has been added in the manuscript to support the susceptibility of horses based on their ACE-2 receptor.

At least, the author need to propose some explanations and conjectures of this infection in discussion part. Does the viral receptor ACE of horse mediate the binding and infection of SARS-CoV-2?

Based on previous work, the authors have shown no association between acute respiratory disease in horses and the detection of SARS-CoV-2 in 667 equids. The literature is sparse concerning reported infections in equids. One experimental study failed to induce infection in a horse, although that study used an ancestral SARS-CoV-2 strain. As shown for other animal species, the interaction between SARS-CoV-2 infected humans (symptomatic and asymptomatic) is required to allow spillover of SARS-CoV-2. This is a rare event in the horse industry compared to human to companion animal interactions (dogs and cats). The authors have recently shown that 35/587 (5.9%) of healthy racehorses with close contact to track personnel testing qPCR-positive for SARS-CoV-2 had detectable antibodies to SARS-CoV-2. In the opinion of the authors, possible spillover from COVID-19 humans to horses is a rare event. Further, various viral and host factors may also contribute to a successful infection. The authors have added information in the discussion regarding explanation of the infection.

Reviewer 2 Report

Please see the comments attached. Thank you!

Author Response

This paper talks about a detection of COVID-19 infection in a horse which is not commonly seen during the pandemic time, and the horse got the virus from an infected horse owner. The paper is essential because it focuses on the hot spot of the virology field since the pandemic is still bringing big impacts to human life all over the world. The paper also has strong significance, since viruses infect and transmit from human to human, human to animals or the way back could be dangerous to public when viruses adapt and mutate. Overall, the finding of this paper is important. 

 I just want to point out that maybe the authors can emphasize the significance of the paper in the discussion part by talking more on the transmission between human and animals. For instance, EEEV infects horses and human; bird flu could also infect human. So, what may happen once virus adapt on another animal and then back on human is a very important point to discuss.

The authors appreciate the reviewer bringing up this point of discussion. Determining the susceptibility of livestock (including horses) to SARS-CoV-2 is essential, considering the negative outcome experienced by farmed mink. The point raised by the reviewer was highlighted at the end of the discussion. While spillover transmission and horse-to-horse transmission is required to promote selection of SARS-CoV-2 with possible reverse zoonosis, there is fortunately little evidence that this situation apply to equids.  

 Also, is there any virus isolated from the infected horse and if so, any differences between the virus from the horse and the virus from the human horse owner?  

Unfortunately, virus was not detected in blood, feces and nasal secretions from the horse that seroconverted. Reasons for the lack of virus detection likely relate to the short infection phase and the weekly interval of sample collection.

 Just curious. Based on the previously published data, why a few horses can get it, but not the majority of the horses? This brings out another question: any hypothesis or guess on what makes this specific horse infected but not the other one?

This is a great question that may be better answered in a review article. Many viral, host and environmental factors determine successful infection. When looking at the two horses, the only difference was age, which has been shown in cats to influence susceptibility (i.e. younger cats are apparently more susceptible to SARS-CoV-2 infection compared to older cats). A sentence regarding age was added in the manuscript.